# Roles for DNA polymerase δ in initiating and terminating leading strand DNA replication

Zhi-Xiong Zhou [1,3], Scott A. Lujan[1,3], Adam B. Burkholder[2], Marta A. Garbacz[1] & Thomas A. Kunkel [1]

Most current evidence indicates that DNA polymerases ε and δ, respectively, perform the bulk of leading and lagging strand replication of the eukaryotic nuclear genome. Given that ribonucleotide and mismatch incorporation rates by these replicases influence somatic and germline patterns of variation, it is important to understand the details and exceptions to this overall division of labor. Using an improved method to map where these replicases incorporate ribonucleotides during replication, here we present evidence that DNA polymerase δ universally participates in initiating leading strand synthesis and that nascent leading strand synthesis switches from Pol ε to Pol δ during replication termination. Ribonucleotide maps from both the budding and fission yeast reveal conservation of these processes. These observations of replisome dynamics provide important insight into the mechanisms of eukaryotic replication and genome maintenance.

[1] Genome Integrity & Structural Biology Laboratory, National Institute of Environmental Health Sciences, NIH, DHHS, Research Triangle Park, NC 27709, USA. [2] Integrative Bioinformatics Support Group, National Institute of Environmental Health Sciences, NIH, DHHS, Research Triangle Park, NC 27709, USA. [3] These authors contributed equally: Zhi-Xiong Zhou, Scott A. Lujan. Correspondence and requests for materials should be addressed to T.A.K. (email: kunkel@niehs.nih.gov)

The division of labor among replicative DNA polymerases (replicases) affects basic biological processes that influence everything from evolution to pathogenesis. Replication of eukaryotic nuclear DNA is initiated when DNA polymerase α (Pol α)-primase synthesizes short RNA–DNA primers that are subsequently extended during synthesis of the two DNA strands (Fig. 1a). Studies of unchallenged DNA replication in yeast systems, using mutations and genomic ribonucleotides as biomarkers of polymerase activity, indicate that Pol δ conducts the majority of discontinuous lagging strand Okazaki fragment synthesis, while Pol ε accomplishes the majority of continuous leading strand DNA replication[1–4]. This canonical model of

**Fig. 1** Models of canonical polymerase division of labor and exceptions at replication origins and termination zones. **a** Replisome components and canonical polymerase division of labor. Red, green, and blue denote Polymerases α, δ, and ε, respectively, or the nascent DNA tracts they synthesize. DNA strands (colored bars) and proteins are not shown to scale. Other replisome components are omitted for simplicity. **b** A model of replication initiation. Replication initiates with Pol α priming on both strands. On the lagging strand, priming is repeated with Pol α passing the 3′ terminus to Pol δ for Okazaki fragment synthesis. On the leading strand, Pol α passes the 3′ terminus to Pol δ, which then catches the receding helicase complex and passes the 3′ terminus to Pol ε. **c** A model of replication termination wherein Pol ε disengages from the 3′ terminus and Pol δ assumes responsibility for the remainder of leading strand synthesis

polymerase division of labor is also supported by physical and biochemical data (refs. [5,6], reviewed in refs. [7,8]).

Replicative DNA polymerases incorporate ribonucleotides into the eukaryotic nuclear genome at remarkably high rates, with approximately one rNMP incorporated for every 600–5000 dNMPs[9,10]. These genomic ribonucleotides are primarily removed by an RNase H2-dependent process known as ribonucleotide excision repair (RER)[9,11]. In cells lacking RER, these ribonucleotides can be used as biomarkers for tracking replicase enzymology across genomes[1,2]. Ribonucleotides in DNA have been mapped using several techniques, including our own Hydrolytic End sequencing (HydEn-seq)[1,2,12,13]. Here we describe an improved version that replaces alkaline hydrolysis of ribonucleotides (now dubbed Alk-HydEn-seq) with enzymatic hydrolysis by *E. coli* RNase HII (RHII-HydEn-seq; Supplementary Fig. 1). We continue to use the general term HydEn-seq to refer to all mapping of polynucleotide termini generated by hydrolysis.

The replicative polymerases were tracked by mapping ribonucleotides in RER-deficient *Saccharomyces cerevisiae* strains using variants of yeast Pols α, δ and ε (e.g. *Pol α-Y869A*, *Pol δ-L612G* and *Pol ε-M644G*, respectively), each of which is more promiscuous for ribonucleotide incorporation than its wild type counterpart. Averaged across efficient replication origins, we recently showed that during initiation of leading strand synthesis, about 180 bp, or roughly one Okazaki fragment, is synthesized by Pol δ between initial priming by Pol α-primase and extensive synthesis by Pol ε[14]. This implies that Pol δ contributes to leading strand synthesis at replication origins (Fig. 1b). This idea is supported by replisome reconstitution studies in vitro with synthetic origins[15]. It remains unclear whether all origins utilize this leading strand initiation mechanism, whether this mechanism is conserved among eukaryotes and whether exceptions to the canonical division of labor among the replicases exist elsewhere in eukaryotic genomes. Here we show prevalent exceptions to the canonical polymerase division of labor at origins, and we suggest an unexpected mechanism of replication termination in eukaryotes.

## Results

**RHII-HydEn-seq.** Genomic ribonucleotides are prone to spontaneous hydrolysis in alkaline condition, resulting in strand breakage at the 3′ of the rNMP. Treating genomic DNA with alkli thus exposes RNA–DNA junctions, specifically the 5′ dNMP and 3′ rNMP with 2′−3′ cyclic phosphate. The 5′ end of the exposed dNMP can be captured by an Illumina platform compatible adaptor and thus analyzed by high-throughput sequencing. In the original Alk-HydEn-seq, 5′ ends were analyzed in genomic DNA of ribonucleotide-promiscuous Pols α, δ and ε strains[1]. Strand bias was interpreted as preferential ribonucleotide incorporation between the two DNA strands, providing a global view of polymerase division of labor during replication. To more quantitatively study local polymerase usage, we now report a modified version of HydEn-seq that we call RHII-HydEn-seq, for mapping ribonucleotides across the genome. The main changes are switching from alkaline hydrolysis to *E. coli* RNase HII digestion, which more specifically cleaves at single embedded ribonucleotides, restriction digestion to introduce internal standards, inclusion of non-treatment control and 5′ end blocking by phosphatase (see the "Methods" section and Supplementary Fig. 1). These modifications were adopted to reduce background noise, increase specificity towards embedded ribonucleotides and improve quantitative comparisons among datasets. Improved genome-wide HydEn-seq maps from RER-deficient strains with either wild type or variant replicases were used to solve a system of equations for the strand-specific contribution of each replicase

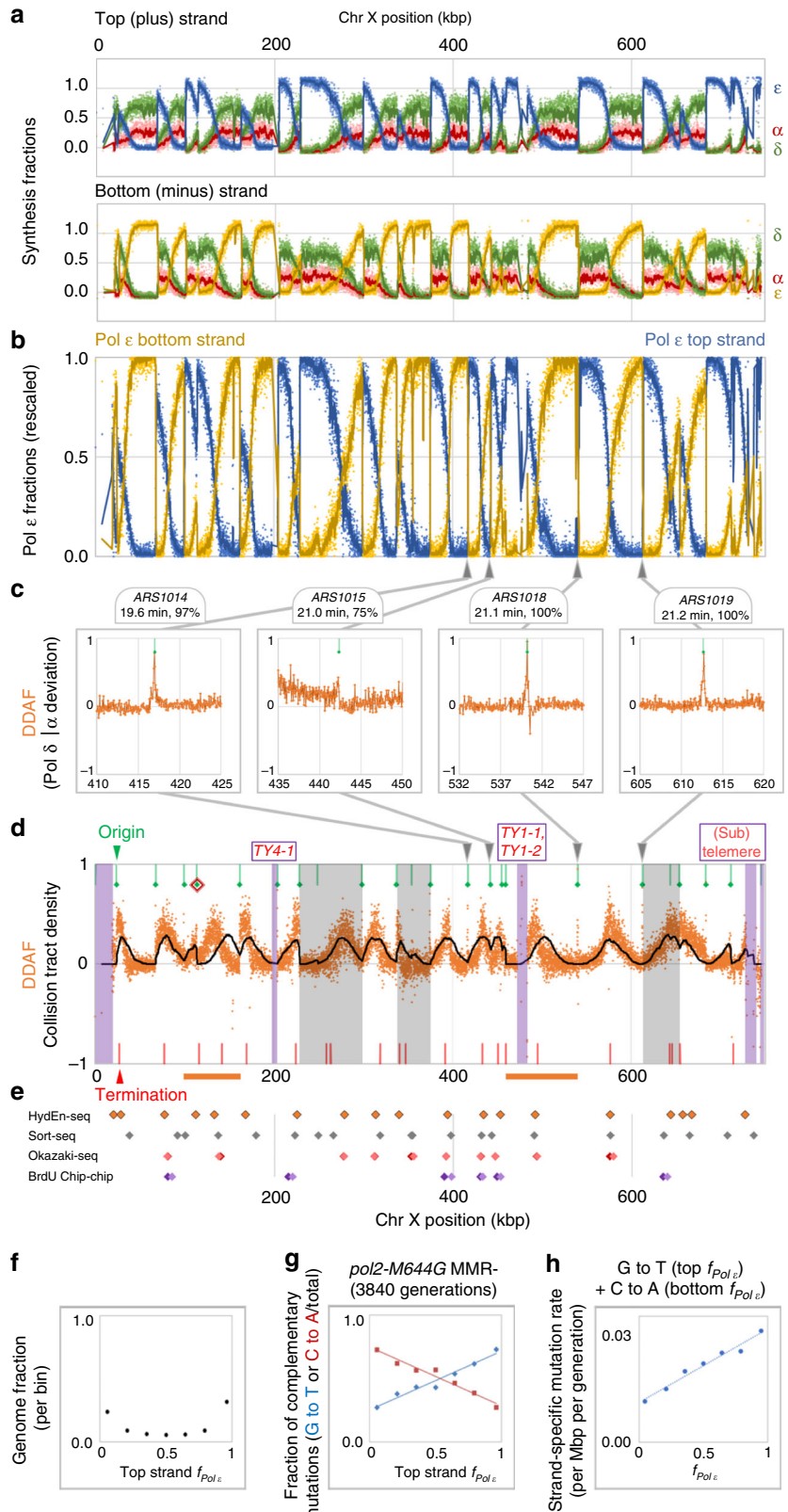

(Fig. 2a, Supplementary Fig. 2 and see the "Methods" section)[14]. The calculations assume that variant replicases synthesizes the same DNA stretches as wild type replicases and that their relative ribonucleotide incorporation rates are constant across S-phase for each polymerase. If deviations from these assumptions are ever discovered, then the models must change accordingly.

**Pol δ conducts a significant portion of leading strand synthesis in *S. cerevisiae*.** We first performed RHII-HydEn-seq in *S. cerevisiae* RER-deficient strains. Replication origins were identified as abrupt shifts in the polymerase synthesis fractions on both strands (Pols ε in Fig. 2b). Importantly, the lower background noise of RHII-HydEn-seq allows higher sensitivity and reveals

**Fig. 2** Polymerase usage across *S. cerevisiae* Chromosome X indicates deviations from a canonical division of labor that occurs at replication origins and termination zones. Data presented are averages of at least three replicates of strains with wild type polymerase or each of the polymerase variants. Points represent values in 50 bp bins. **a**, **b** Curves are 1 kb moving averages. **a** Fractional strand-specific Pol δ, α, and ε usage (green, red, and blue/yellow, respectively). Noise in RHII-HydEn-seq data push curves slightly outside of the 0–1 range. Steep transitions indicate either active origin positions or low coverage regions (e.g. transposons, telomeres, etc.). **b** Fractional Pol ε synthesis ($f_ε$) of top (blue) and bottom (yellow) strands (1.14x linear rescale to 100% maximum). **c** and **d** The DDAF (orange points) the measurement of division of labor between Pol ε and Pols δ and α. Green bars represent origin positions, and green diamonds indicate those with established firing times (not inferred herein). Red bars indicate predicted collision positions assuming optimal global fork rates and 100% origin efficiency. **c** Exemplar DDAF peaks at four early-firing, relatively efficient replication origins. Curves are unsmoothed. **d** The Chr X DDAF and Monte Carlo simulated fork collision density fit thereto (black curve; 1000 simulations; see the "Methods" section and Fig. S4 for parameters). Note noise around non-unique positions like sub/telomeres and transposons (purple). Orange bars below the horizontal axis indicate inter-origin tracts where simulation and observation deviate. The red diamond indicates an origin where simulated collision peaks are closer than DDAF peaks in both directions, suggesting later firing than expected. Gray backgrounds indicate tracts adjacent to origins with firing times inferred herein, rather than previously measured[20]. **e** Comparing DDAF peak positions (orange with black border) with replication termination peak positions measured from BrdU ChIP-chip (light/dark purple for left/right terminus[22]) or calculated from Okazaki fragment sequencing (red/pink for replicate A/B[21]) or sort-seq (gray; high confidence, i.e. score > 0.004[20]). **f–h** Mutation rates as orthogonal confirmation of $f_ε$ calculations. Mutations accumulated over 3840 generations in mismatch repair-deficient *S. cerevisiae* with mutator Pol ε variant M644G[18]. **f** Fractions of the *S. cerevisiae* genome partitioned (bin) by top strand $f_ε$. **g** The fraction of G to T substitutions increases linearly with top strand $f_ε$. The opposite holds for C to A substitutions. **h** Rates of Pol ε characteristic mutations increase with $f_ε$. Strandedness was assigned given the preference of M644G Pol ε for making C-dT (template C; incoming dTTP) vs. G-dA mispairs

finer details, including numerous inefficient/late origins, thereby expanding the list of active origins identified by HydEn-seq from 394[1] to 465 (Supplementary Data 1), 11 of which were not found in the OriDB[16]. Overall, these improvements produced a fine-grained map of polymerase usage that allowed us for the first time to examine the action of replicative polymerases at individual origins, and also in individual inter-origin tracts.

To probe the global division of labor between the canonical leading and lagging strand polymerases, we compared the fraction of top strand synthesis due to Pol ε, from RHII-HydEn-seq (Supplementary Fig. 2i: orange; $f_{i,ε}$), to the estimated fraction of the top strand replicated as the nascent leading strand (Supplementary Fig. 2i: blue; $F_{i\ \text{lead,top}}$). The formulae for the latter assume the canonical division of labor, as previously described[17]. Across the genome, top strand Pol ε synthesis fractions ($f_{i,ε}$) were lower than top strand leading strand fractions ($F_{i\ \text{lead,top}}$), especially where $F_{i\ \text{lead,top}}$ approached 50%, i.e. at origins and in termination zones (Supplementary Fig. 2j). This suggests less synthesis by Pol ε in those regions than would be expected from the canonical division of replicase labor.

To probe the local division of labor, we calculated the deviation from the expected Pol δ and Pol α fraction of synthesis (DDAF) for each 50 base pair genomic bin (Fig. 2c, d). Hypothetically, the canonical division of labor would yield a DDAF score of zero, local synthesis of both strands by Pols δ and α would yield a DDAF of 1, and local synthesis of both strands by Pol ε would yield a DDAF of −1. Surprisingly, we observed numerous DDAF peaks with positive values (Fig. 2c, d), indicating that a significant portion of leading strand synthesis is accomplished without Pol ε. The average DDAF across the genome suggests that Pol δ is responsible for 18% of leading strand synthesis in *S. cerevisiae*. This percentage of leading strand replication by Pol δ is consistent with earlier studies indicating that Pol ε likely performs the majority but not all of leading strand replication[7].

**Correlation of RHII-HydEn-seq maps with genome mutation data**. In addition to increased ribonucleotide permissiveness, the polymerase variants also create characteristic mutation signatures during DNA synthesis. To verify the calculated fractional polymerase usage, we compared Pol ε synthesis fractions to mutation rates in yeast strains with *Pol ε-M644G*, which is also prone to misincorporate base substitutions across the genome[18]. If mutator polymerase error rates and ribonucleotide incorporation rates are constant regardless of origin proximity, then fractional

polymerase use should correlate with the distribution of their characteristic mutation signatures. For instance, mutator polymerase variants *Pol ε-M644G*, *Pol α-L868M*, and *Pol δ-L612M* preferentially make C-to-A transversion substitutions through C-dT mispairs on one strand rather than through complementary G-dA mispairs on the other strand[18]. This bias is clear when the fraction of G to T and C to A mutations are plotted vs. the top strand Pol ε synthesis fraction in a DNA mismatch repair (MMR)-deficient strain containing *Pol ε-M644G* (Fig. 2f–h). As predicted, characteristic substitution rates from *Pol ε-M644G* strain[18,19] are linearly correlated with Pol ε synthesis fractions for the example above (Fig. 2h, $R^2 = 0.973$, $n = 1915$). The same is true for other *Pol ε-M644G* mutation types and for the characteristic substitutions rates in strains *Pol α-L868M* or *Pol δ-L612M* ($R^2 > 0.95$, when $n > 300$ and bias >2×; Supplementary Fig. 4d). Thus, mutation data provide orthogonal confirmation for the polymerase division of labor calculated from the RHII-HydEn-seq maps. The density of the current mutation data collection, however, is not sufficient to study local exceptions to the division of labor discussed below.

**Pol δ is ubiquitous in *S. cerevisiae* leading strand initiation**. The DDAF peaks are of two types, narrow peaks that coincide with efficient replication origins (Fig. 2c) and broad peaks elsewhere (Fig. 2d and Supplementary Fig. 3). Narrow DDAF peaks are found at replication origins (Figs. 2c and 3a). They are lower for inefficient origins (Fig. 3b) than for efficient origins (Fig. 3c). After baseline subtraction, peak areas vary linearly with origin efficiency (Fig. 3d; $R^2 = 0.986$). This implies that polymerase usage is uniform across origins. From this linear relationship, the area under a hypothetical origin with perfect efficiency is 177 bp, approximately one Okazaki fragment. This is consistent with our previous meta-analysis of efficient origins[14], and strongly indicates that Pol α-to-δ-to-ε leading strand initiation is essentially universal across the genome in unstressed *S. cerevisiae* cells.

**The role of Pol δ in leading strand synthesis during replication termination**. The broad DDAF peaks between origins are widespread, being found between nearly all pairs of adjacent origins (Figs. 2d and 3e). They have roughly Gaussian or truncated Gaussian shapes (Fig. 3f, g and Supplementary Fig. 5), but their areas are not correlated with the efficiency of adjacent origins (Fig. 3h). We suggest that these peaks represent a switch from Pol

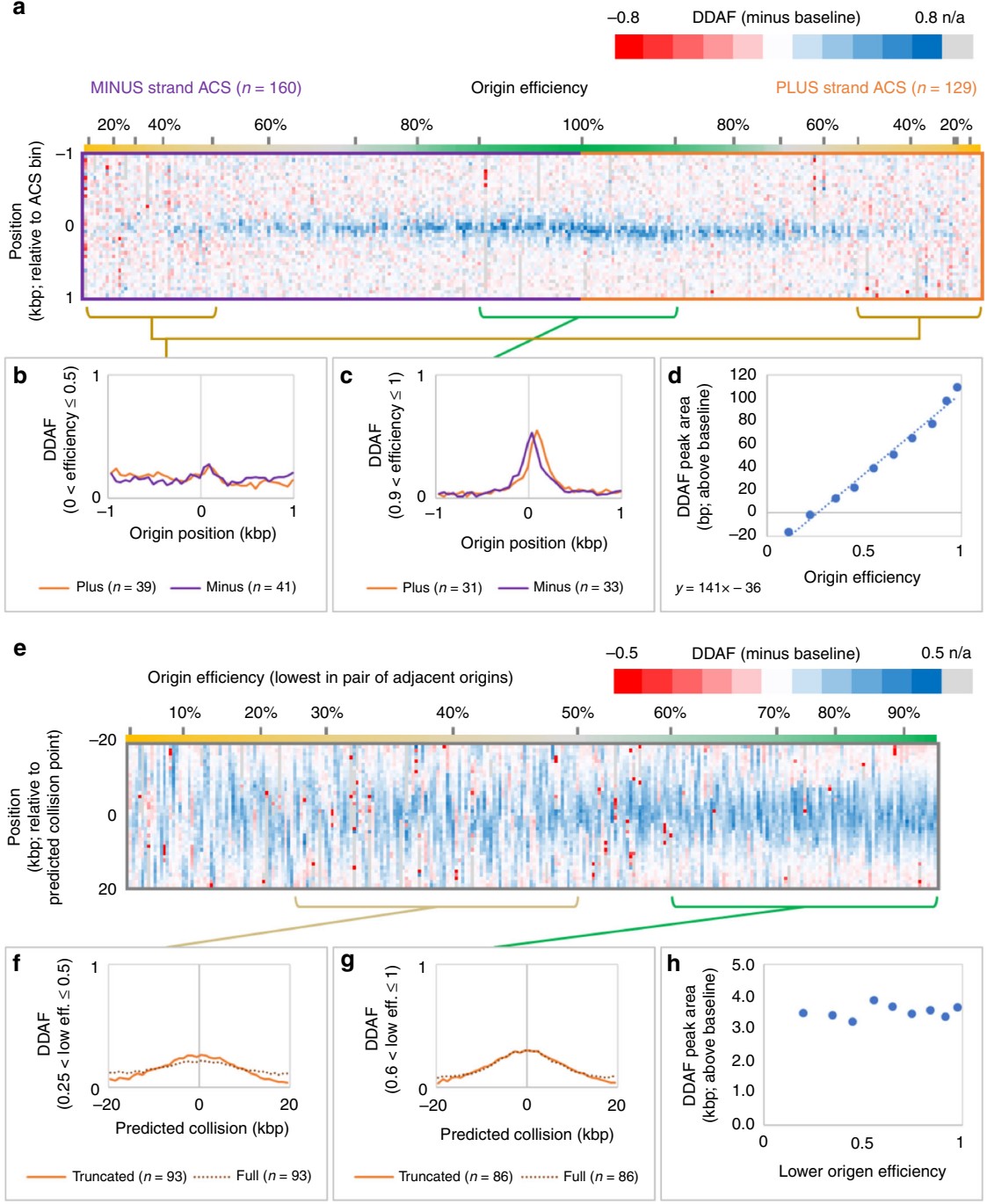

**Fig. 3** DDAF peaks at replication origins and termination zones in *S. cerevisiae*. **a** A DDAF heatmap in 50 bp bins for one kilobase on either side of 289 replication origins (green bars in Figs. 1d and S2). Origin motifs (ACS motifs) on the plus (top) strand are grouped as "plus strand ACS" and minus (bottom) strand oriented origins are gouped as "minus strand ACS". Origins are ranked by efficiency. Blue indicates a high DDAF value (less Pol ε usage), red denotes the opposite. **b** Averaged DDAF value at inefficient origins (efficiency < 0.5). Orange curves are for plus strand origins and purple curves are for minus strand origins. The number of origins used is indicated. **c** Averaged DDAF value at efficient origins (efficiency > 0.9). **d** DDAF peak areas (after baseline subtraction) increase linearly with origin efficiency ($R^2 = 0.986$). The negative value at origin efficiency of zero is due to aggressive background subtraction (see the "Methods" section). **e** As per **a** but in 1000 bp bins for 20 kb on either side of predicted collision points (red bars in Figs. 1d and S2) for forks proceeding from 259 well-separated adjacent origins (distance ≥ 20 kb). The heatmap is ranked by the efficiency of the lesser ones of the pairs of adjacent origins. On average, origin pairs with a lesser member of both **f** moderate and **g** high efficiency have broad DDAF peaks centered at predicted termination zones, both if the data is truncated at the member origins (orange curves; heatmap in Fig. S4) or not (brown dotted curves). **h** The areas under truncated curves are independent of flanking origin efficiency. Slight negative areas for some inefficient origins suggest that true baselines are somewhat lower

ε to Pols δ and α on the leading strand at termination zones, where two forks moving in opposite directions converge. A Monte Carlo simulation of converging forks models these peaks well (Fig. 2d and Supplementary Fig. 3, black line; Supplementary Figs. 6 and 7 and see the "Methods" section for fitting parameters). Moreover, the DDAF peaks largely colocalize with three published lists of termination zones/peaks (Fig. 2e)[20–22], strongly suggesting that the broad DDAF peaks in termination zones are due to Pol δ synthesis of both DNA strands as forks converge.

Assuming that the broad DDAF peaks correlate with fork collision frequency, then they represent a map of DNA replication termination zones that is both comprehensive and agnostic to origin positions, firing times and fork velocities (Supplementary Fig. 3). In future studies, such agnosticism might be important when mapping termination zones in organisms, where these parameters are unknown, imprecise, or highly variable, such as the diffuse origins of vertebrates[23]. This imprecision is seen here in budding yeast, where 41 of the 205 tracts between origins with preset positions and firing times (Fig. 2d and Supplementary Fig. 3, white regions) have some obvious deviation (e.g. Fig. 2d, orange bars) between simulated collision frequencies and observed DDAF peaks (Supplementary Fig. 3, black line and orange points, respectively). Most of these deviations are minor. Twenty-three of these deviations are in symmetrical sets, as expected where actual firing times differ from listed times. For example, if the actual firing time is later than the listed time for a given origin, then approaching forks have more time to converge on that origin before termination. Thus, both adjacent termination peaks would be closer than expected to that origin (e.g. Fig. 2d, red diamond). The remaining deviations may indicate other determinants of replication termination, such as chromatin state or locally variable fork velocity[5,24].

**RHII-HydEn-seq in Schizosaccharomyces pombe.** Our S. cerevisiae data reveal that Pol δ synthesizes both strands during replication initiation and termination. To explore the evolutionary conservation of this mechanism, we turned to another model organism, the fission yeast S. pombe. The ancestors of S. pombe and S. cerevisiae lines diverged about 500 million years ago[25]. Though they have genomes of similar size, S. pombe seems to have evolved more slowly than S. cerevisiae, and is more similar to their common ancestor and therefore to the common ancestor of all opisthokonts (fungus-like and animal-like organisms)[26]. In S. pombe, replication origins are not defined by specific consensus sequences, but rather are found at prevalent intergenic AT-rich sequences[27,28], as has been suggested for metazoans[29,30].

Using PU-seq, an approach similar to HydEn-seq, it was previously demonstrated that mapping ribonucleotide incorporation is an efficient way to track polymerase usage in S. pombe[2]. Thus, we applied RHII-HydEn-seq procedure to RER-deficient S. pombe strains with wild type polymerases or ribonucleotide-permissive Pol ε or Pol δ variants (cdc20-M630F and cdc6-L592G, respectively; Fig. 4a). Sites of replication initiation, again defined by abrupt shift in Pol ε synthesis fractions, are broader in S. pombe than in S. cerevisiae. The PU-seq study identified 1145 origins in S. pombe[2], a majority (906 origins, 79%) with efficiencies below 50% and only six with efficiencies over 80%[2]. Efficient origins in S.pombe are supported by Pol ε synthesis fraction shifts calculated from RHII-HydEn-seq data. The sum of origin efficiencies, approximating the average number of origins that fire in an S. pombe S-phase, is 295. While this value is similar to the average of 265 origins found with our S. cerevisiae Monte Carlo simulation, the greater density of inefficient origins in S. pombe results in many low but still sharp transitions in

polymerase usage plots (Fig. 4a). Many such transitions occur at predetermined origins, but many others do not. This suggests a multitude of uncatalogued origins. Indeed, other studies have identified hundreds of additional S. pombe origins and even suggest that most AT-rich sequences serve as initiation sites with some efficiency[27,28]. The net result is a more complex and visually noisier map of polymerase usage.

**Pol δ participation in leading strand synthesis during initiation and termination in S. pombe.** S. pombe DDAF plots are also more complex than their S. cerevisiae equivalents (Fig. 4b, c). DDAF peaks are nonetheless visually detectable at efficient origins (Fig. 4b). Thus, we aligned PU-seq-derived origins[2] and found that identified origins mostly center around DDAF peaks (Fig. 4e, f). As in S. cerevisiae, the area under the peaks correlated linearly with origin efficiency (Fig. 4g). Therefore, there is agreement between the two ribonucleotide mapping techniques (PU-seq and RHII-HydEn-seq). The peak area implies that Pol δ synthesizes 425 bp of leading strand during an average origin initiation event, two to three times as much as in S. cerevisiae.

Next, we examined polymerase usage at S. pombe termination zones. Incomplete knowledge of origin counts, positions, and firing times precludes a detailed Monte Carlo simulation of for collision densities. However, on average, the DDAF does peak between relatively efficient (>0.2), well-spaced (>20 kb), early firing origins with no known intervening origins of significant efficiency (>0.1) (Fig. 4h, orange curve), similar to termination peaks observed in S. cerevisiae (Fig. 4h, brown dashed curve). The areas of peaks at origins and termination zones are also similar in the two species (Fig. 4i; corrected by background subtraction, see the "Methods" section). These results in S. pombe suggest an evolutionarily conserved paucity of Pol ε on the leading strand during replication initiation and termination.

**Discussion**
The discovery of the Pol δ participation on both strands at replication origins presents a novel model of replication initiation[5,14,15]. In contrast to canonical view that Pol α directly primes both leading and lagging strand synthesis, this model suggests that each leading strand is primed on the opposite side of the origin, by an Okazaki fragment synthesized by Pols α and δ (Fig. 1b). Despite support for this model from in vitro reconstitution experiments and from polymerase usage maps averaged across many efficient origins, the predominance of this mechanism was an open question. Here we use detailed maps of polymerase usage in S. cerevisiae and S. pombe to show that Pol δ participates in leading strand initiation across eukaryotic genomes, regardless of origin efficiency. Interestingly, on average, the Pol δ tracts at S. pombe origins are somewhat longer than the average eukaryotic Okazaki fragment (Fig. 4i). This may be related to the wider nucleosome-depleted region at S. pombe origins[31].

It was previously proposed that when forks converge during termination, the leading strands pass each other and catch up with the downstream lagging strands[32,33] (Supplementary Fig. 8). However, our data suggest a switch from Pol ε to Pol δ on the leading strands during replication termination. Thus, converging replisomes must undergo some reconfiguration or disassembly before their encounter in order to allow Pol δ to take over synthesis on both strands (Fig. 1c). It is unclear what effect this transition on leading strand will have on lagging strand synthesis. Depending on the degree of Pol α involvement, we consider four possible scenarios for resolving impending fork collisions (Supplementary Fig. 8). For reasons discussed therein, we favor a

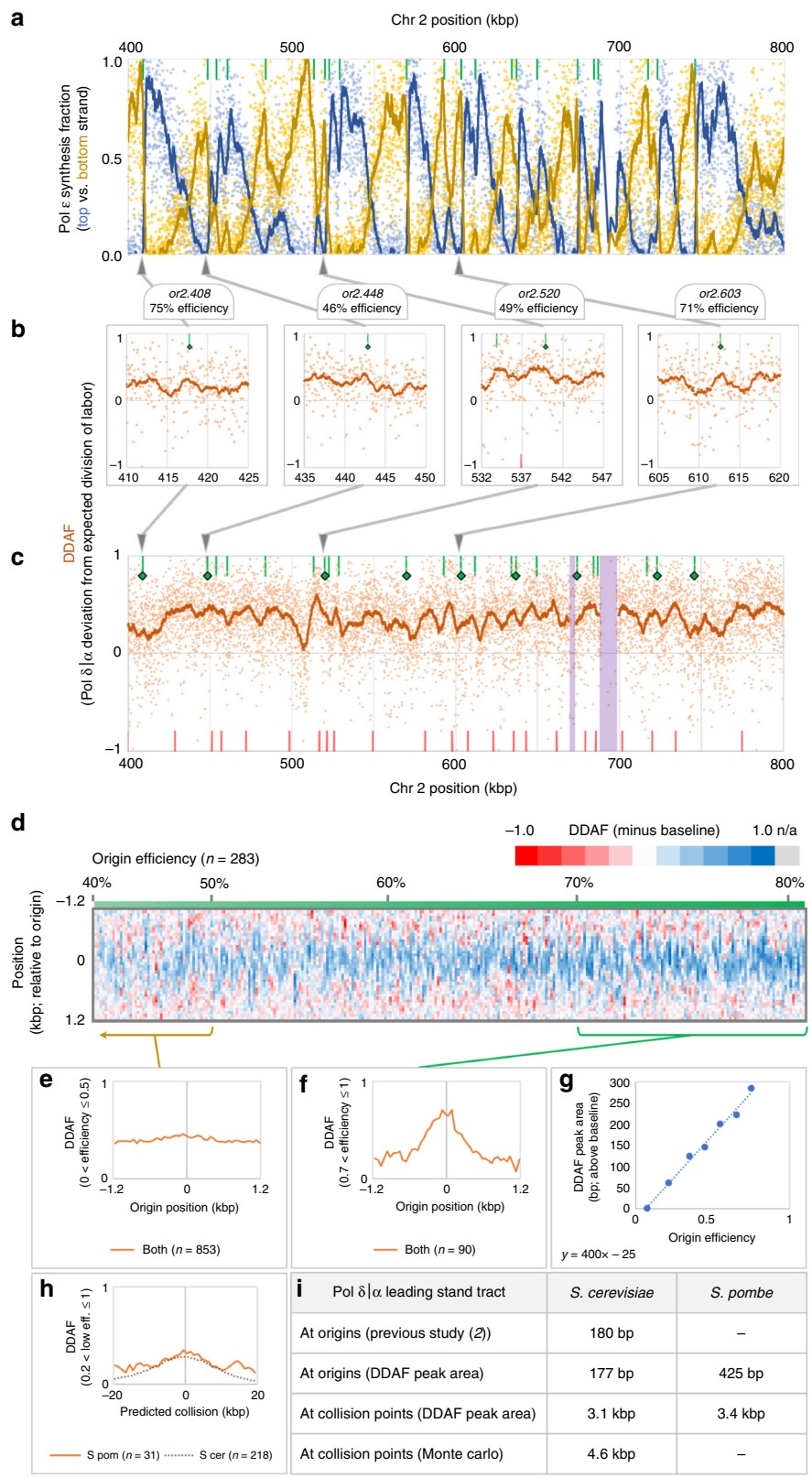

model, where Pol δ takes over leading strand synthesis, while Okazaki fragment synthesis on the lagging strand remains unperturbed until forks collide and remaining gaps are filled by Pol δ (Supplementary Fig. 8d).

The switch from Pol ε to Pol δ on the leading strand may be a response to challenges during termination, such as accumulation of topological stress as forks converge. Topological stress arising from strand unwinding complicates replication[32]. As supercoils accumulate, parental strand stress can be relieved through rotation of the incoming forks, forming precatenanes that are resolved by topoisomerases[32]. However, as forks converge, supercoiling may build faster than it is relieved, which would

**Fig. 4** Exceptions to a canonical division of labor are conserved in *S. pombe*. Data are an average of at least three replicates for each genotype. **a** Fractional Pol ε synthesis ($f_{Pol\ ε}$) of top (blue) and bottom (yellow) strands. Points represent 50 bp bins. Curves are 2 kb moving averages. **b, c** The DDAF profile (orange points): the total fraction of synthesis by Pols α and δ across both strands. Green bars represent origin positions. Green diamonds indicate origin efficiencies over 0.4. Red bars indicate predicted collision positions assuming uniform global fork rates, uniform firing times and 100% origin efficiency. Points represent 50 bp bins. Curves are 3 kb moving averages. **b** DDAF peaks at four relatively efficient replication origins. Curves are 1.5 kb moving averages. **c** The DDAF across a section of Chromosome 2. Curves are 4 kb moving averages. Non-unique regions are excluded (purple). **d** A DDAF heatmap in 50 bp bins (3-bin moving average) for 1.2 kb on either side of 283 replication origins (green bars in **a**). Blue indicates a high DDAF (less Pol ε usage), red denotes the opposite. **e** Averaged DDAF curve at inefficient (efficiency < 0.5). The number of origins used is indicated below the graph. **f** Averaged DDAF peak at efficient (>0.7) origins. **g** DDAF peak areas (after baseline subtraction) increase linearly with origin efficiency ($R^2 = 0.990$). **h** Early firing origin[42] pairs (efficiencies > 0.2, intervening origin efficiency < 0.1, separation > 20 kb) have broad DDAF peaks centered near the inter-origin midpoint (orange curve; truncated at member origins). These peaks resemble collision point-centered *S. cerevisiae* DDAF peaks (brown dotted curve). **i** Comparison of tract length inferred from DDAF peak area in *S. cerevisiae* and *S. pombe*

challenge fork progression. During SV40 replication, forks slow or pause as they converge, leaving long lived gaps[34,35]. In the *S. cerevisiae*, BrdU incorporation in synchronized cultures revealed convergent fork pausing at a termination zones (TERs) (e.g. Fig. 2e, purple) with an average separation of 5.1 kb[22]. Even though this study was conducted in yeast cultures at low temperature or under treatment of hydroxyurea, most of the reported TERs are supported by our HydEn-seq data. Moreover, the average length of TERs is similar to the 4.6 kb Pol δ leading strand tracts found during optimization of our Monte Carlo simulation to DDAF (Fig. 4i). Analogous stalling was not observed in plasmid-based assays in *Xenopus* egg extracts[33], reconstituted DNA replication systems or yeast extracts[36]. However, it is possible that the limited size of the plasmids may preclude such experiments from recapitulating certain characteristics of chromosomal replication. If such stalling occurs universally in eukaryotic cells, replisomes may disassemble or reorganize without active Pol ε. In the *S. pombe*, Pol δ replicates both strands after fork restart[37]. The fork pausing may be either the signal for or the consequence of such replisome reorganization. Because Pol ε is tightly bound to the CMG helicase complex[38], such reorganization would allow additional rotational freedom around the leading strand template DNA backbone between the helicase and Pol δ and thus potential relief of some topological stress. Moreover, the remodeling of the replication fork might help prevent steric clash between converging forks.

It is worth noting that some of the polymerase variants used in this study show signs of slowed S phase progression[9,39]. Thus, the in vivo behavior of each polymerase variant may be different from their WT counterpart in an unknown way. However, in each of these strains, two replicative polymerases are wild-type (e.g. in pol2-M644G strain, Pol α and Pol δ are wild-type). Since our calculations use data from all three polymerase variants (see the "Methods" section and Supplementary Fig. 2), the resulting polymerase usage profile should resist the potential swaying effect of any single mutant. Nonetheless, the possibility still exists and if true, the swaying effect is more likely to influence certain quantitation, such as the tract length of termination zones, rather than the overall conclusions.

Overall, this study confirms that Pol δ is an intermediary between Pols α and ε during leading strand initiation, and it uncovers an important role for Pol δ during replication termination. These findings reveal the intrinsic plasticity of DNA polymerase usage during DNA replication. This plasticity may help the replication machinery accommodate and overcome different challenges during genome duplication. Similar DNA polymerase dynamics in the two distantly related yeast models, and known similarities between *S. pombe* and metazoan DNA replication, suggest that the underlying mechanisms are likely to be conserved in higher eukaryotes.

## Methods

**Yeast strains.** The RER-deficient *S. cerevisiae* strains were derived from Δ|(−2)|-7B-YUNI300 (*MATa CAN1 his7-2 leu2Δ::kanMX ura3Δ trp1-289 ade2-1 lys2ΔGG2899-2900*) and were previously described[1]. The *S. pombe* strains used in this study are from the previous study[2].

**RHII-HydEn-seq library construction.** RHII-HydEn-seq libraries are based on Alk-HydEn-seq previously described[1,14] with modifications by the following steps (Supplementary Fig. 1, primers listed in Supplementary Table 1). Briefly, extract yeast genomic DNA (gDNA) using Epicenter MasterPure Yeast DNA Purification Kit. Treat 200 ng gDNA with 1 unit of Shrimp alkaline phosphatase (rSAP, New England Biolabs) for 30 min at 37 °C. Denature rSAP and restriction digest with 20 units of SbfI-HF at 37 °C for 1 h. Split the DNA into two fractions. Treat one fraction with 10 units of RNase HII (NEB) at 37 °C for 2 h and designate the other fraction as non-treatment control. Denature the DNA by incubation at 90 °C for 2 min and ligate to adaptor ARC140 by 10 units of T4 RNA ligase 1 (NEB) overnight at 25 °C. Denature DNA and anneal to the duplex adaptor ARC76/77. Synthesize the second strand using 4 units of T7 DNA polymerase (NEB). Amplify the library for 20 cycles by primer ARC49 and an indexing primer using KAPA HiFi HotStart Ready Mix (KAPA Biosystems). 0.8 volumes of MagBio HighPrep PCR beads were used to purify DNA between steps that require changing buffers and/or removal of oligo adaptors and in the final library cleanup. The library was sequenced on an Illumina HiSeq 4000 machine for paired-end 50 bp reads. The rSAP, SfbI-HF, and RNase HII treatments are the main modifications to the original protocol to improve on signal-to-noise ratio, quantitation and specificity.

**Data processing, alignment, and normalization.** RHII-HydEn-seq reads were processed and aligned as previously described[1]. Briefly, paired or single reads that align to the L03 reference genome[18,40] allowing one mismatch were retained. The geometric means of end counts mapped to SbfI sites across all samples were determined and used for determination of normalization factor. Each sample was normalized by the normalization factor determined as median of the ratios of all SbfI sites to their respective geometric means. SbfI sites that have zero counts in one or more samples were excluded from the analysis. SbfI sites were used for normalization instead of total mapped reads because the restriction sites are not influenced by ribonucleotide incorporation and thus are better internal standards.

**Calculating the division of polymerase labor at replication origins.** The normalized RHII-HydEn-seq end count (see above) at position $i$ on strand $j$, for each data set $K$, is background subtracted to give the corrected end count ($y_{i,jK}$; e.g. Supplementary Fig. 2a, b), as reported previously[14]. There are three exceptions to the previous method. First, the previous study treated all replicates independently, whereas in this study replicate data sets were averaged, before background subtraction and after normalization with restriction endonuclease end counts. Second, background subtraction used RHII-HydEn-seq data from DNA samples that were not treated with RNase HII. Third, ribonucleotide incorporation rates ($s_k$) for each polymerase ($k$) were not found through least-squares fitting. Instead they were extracted from genomic windows that were synthesized by forks progressing from a single origin at least 90% of the time. These windows were found using a published method for estimating nascent strandedness[17] (e.g. Supplementary Fig. 2c, d). These nascent strandedness estimates are only accurate where the canonical division of labor holds, as expected in these windows. For example, where the top strand is always the nascent leading strand, the end density should be equivalent to the ribonucleotide incorporation rate of the Pol ε variant. Note that these windows do not overlap replication origins or termination zones. The corrected end counts were rescaled slightly such that windows with wild type polymerases in variant strains matched the corresponding windows in the wild type strain. Because 90% is <100%, this estimation will stretch polymerase usage fractions outside of the natural 0–1 range. See the next "Methods" section for correction procedures.

Briefly, as per Garbacz et al.[14], to calculate the fraction of synthesis for polymerase $k$ at position $i$ on strand $j$ ($f_{i,j,k}$), first assume that variant and wild type

polymerases synthesize the same DNA with the same frequency, as per

$$f_{i,j,\text{Pol wild type}} = f_{i,j,\text{Pol variant}} \tag{1}$$

Then make the simplifying assumption that the three replicases are the only means for DNA production, and therefore

$$1 = f_{i,j,\alpha} + f_{i,j,\varepsilon} + f_{i,j,\delta} \tag{2}$$

Given a multiplicative noise factor ($w$; dependent on position and strand but independent of polymerase background[17]; e.g. Supplementary Fig. 2e) and the ribonucleotide incorporation rate of each polymerase $k$ ($s_k$), then the HydEn-seq ribonucleotide density ($y$) in strain $K$ (which has polymerase variant $k$), at position $i$ on strand $j$, is

$$y_{i,j,K} = w_{i,j} \sum_{k=1}^{n} s_k f_{i,j,k} \tag{3}$$

Therefore,

$$w_{i,j} = y_{i,j,\varepsilon MG} / \left( s_\alpha f_{i,j,\alpha} + s_{\varepsilon MG} f_{i,j,\varepsilon} + s_\delta f_{i,j,\delta} \right) \tag{4}$$

$$w_{i,j} = y_{i,j,\delta LG} / \left( s_\alpha f_{i,j,\alpha} + s_\varepsilon f_{i,j,\varepsilon} + s_{\delta LG} f_{i,j,\delta} \right) \tag{5}$$

and

$$w_{i,j} = y_{i,j,\alpha YA} / \left( s_\alpha f_{i,j,\alpha YA} + s_\varepsilon f_{i,j,\varepsilon} + s_\delta f_{i,j,\delta} \right) \tag{6}$$

Make the simplifying assumption that the three replicases are the only means for DNA production, and therefore

$$f_{i,j,\varepsilon} = 1 - f_{i,j,\alpha} - f_{i,j,\delta} \tag{7}$$

Substitute Eq. (7) into Eqs. (4) and (5) to remove $f_{i,j,\varepsilon}$

$$w_{i,j} = y_{i,j,\varepsilon MG} / \left( s_\alpha f_{i,j,\alpha} + s_{\varepsilon MG} \left( 1 - f_{i,j,\alpha} - f_{i,j,\delta} \right) + s_\delta f_{i,j,\delta} \right) \tag{8}$$

$$w_{i,j} = y_{i,j,\delta LG} / \left( s_\alpha f_{i,j,\alpha} + s_\varepsilon \left( 1 - f_{i,j,\alpha} - f_{i,j,\delta} \right) + s_{\delta LG} f_{i,j,\delta} \right) \tag{9}$$

$$w_{i,j} = y_{i,j,\alpha YA} / \left( s_\alpha f_{i,j,\alpha YA} + s_\varepsilon \left( 1 - f_{i,j,\alpha} - f_{i,j,\delta} \right) + s_\delta f_{i,j,\delta} \right) \tag{10}$$

Substitute Eq. (9) into Eq. (8), to remove $w_{i,j}$, and solve for $f_{i,j,\alpha}$ in terms of $f_{i,j,\delta}$, then repeat this procedure with Eqs. (10) and (8), to produce

$$f_{i,j,\alpha} = \frac{f_{i,j,\delta} \left( y_{i,j,\delta LG}(s_\delta - s_{\varepsilon MG}) - y_{i,j,\varepsilon MG}(s_{\delta LG} - s_\varepsilon) \right) + y_{i,j,\delta LG} s_{\varepsilon MG} - y_{i,j,\varepsilon MG} s_\varepsilon}{y_{i,j,\varepsilon MG}(s_\alpha - s_\varepsilon) - y_{i,j,\delta LG}(s_\alpha - s_{\varepsilon MG})} = \frac{f_{i,j,\delta} A + B}{C} \tag{11}$$

and

$$f_{i,j,\alpha} = \frac{f_{i,j,\delta} \left( y_{i,j,\alpha YA}(s_\delta - s_{\varepsilon MG}) - y_{i,j,\varepsilon MG}(s_\delta - s_\varepsilon) \right) + y_{i,j,\alpha YA} s_{\varepsilon MG} - y_{i,j,\varepsilon MG} s_\varepsilon}{y_{i,j,\varepsilon MG}(s_{\alpha LM} - s_\varepsilon) - y_{i,j,\alpha YA}(s_\alpha - s_{\varepsilon MG})} = \frac{f_{i,j,\delta} D + E}{F} \tag{12}$$

Setting Eq. (11) equal to Eq. (12), then solving for $f_{i,j,\delta}$ yields

$$f_{i,j,\delta} = \frac{CE - FB}{FA - CD} \tag{13}$$

This puts the fraction of synthesis by Pol δ into terms of HydEn-seq end density alone (parameters $y_{i,j,K}$ and $s_k$). Once $f_{i,j,\delta}$ has been computed, solve for $f_{i,j,\alpha}$ by back substituting into Eq. (11) or (12) and then for $f_{i,j,\varepsilon}$ by back substituting $f_{i,j,\delta}$ and $f_{i,j,\alpha}$ into Eq. (7). The multiplicative noise factor may be computed by substituting the synthesis fractions into any of Eqs. (4) through (6). Example results are shown in Supplementary Fig. 2f, g. Calculated $f_{i,j,\varepsilon}$ values deviated from previous leading-strandedness estimates only at origins and in termination zones (Supplementary Fig. 2h, i).

**Calculating the DDAF.** The total fraction of synthesis at position $i$ on both strands by DNA Polymerase ε is (from Eq. (2)) is

$$f_{i,\text{both},\varepsilon} = \frac{\left( f_{i,\text{top},\varepsilon} + f_{i,\text{bottom},\varepsilon} \right)}{\sum_{j=1}^{2} \sum_{k=1}^{3} f_{i,j,k}} = \left( f_{i,\text{top},\varepsilon} + f_{i,\text{bottom},\varepsilon} \right) / 2 \tag{14}$$

Therefore, we define the deviation from expected Pol δ and α fraction of synthesis (DDAF), i.e. the fraction of synthesis not due to Pol ε, as

$$\text{DDAF} = 1 - 2 f_{i,\text{both},\varepsilon} = 1 - f_{i,\text{top},\varepsilon} - f_{i,\text{bottom},\varepsilon} \tag{15}$$

Example results are shown in Supplementary Fig. 2h. Note that $f_{i,\text{top},\varepsilon}$ and $f_{i,\text{bottom},\varepsilon}$ may need to be rescaled to correct for $s_k$ underestimation, as was the case for *Saccharomyces* HydEn-seq data presented herein (divided by 1.14). Such linear rescaling changes neither the positions nor shapes of DDAF peaks but care must be taken to ensure that peak heights and areas are correct. Here we chose the scale factor that set maximum smoothed Pol ε fractions to 1. This choice was validated when the maximum area under DDAF peaks at origins was found to be 177 bp

(calculations below), very close to the 180 bp of the leading strand at origins previously found to be synthesized by Pol δ[14].

**Estimating synthesis tract lengths from DDAF peaks.** Pol α|δ synthesis tract lengths were estimated as the area under DDAF peaks after baseline subtraction. The baselines were estimated from the 10% to 20% of bins at extreme $x$-values in Figs. 3a–c, e–g, 4d–h. This overly aggressive subtraction was used in the absence of a true baseline. For origins, over-subtraction was corrected by linear fitting area vs. efficiency data (Figs. 3d, 4g; area = slope*efficiency + $y$-intercept) and then estimating the maximum peak area with ideal background subtraction as slope*1 + $y$-intercept.

**Monte Carlo simulation of fork collision zones.** Best fits between the Monte Carlo simulation and DDAFs were found based on lists of *S. cerevisiae* origin positions and scaled ($s_t$) firing times derived from the *S. cerevisiae* Origin Database (OriDB)[16,18]. Firing times were measured either from α-factor arrest release[18,41] (Supplementary Fig. 6a, $s_t = 1$), or rescaled to *cdc7-1* arrest release (Supplementary Fig. 6b, $s_t = 3.75$)[16]. The underlying approach simulated origin firing and collision tract position based on three global tunable parameters: fork velocity ($v_f$; 6 and 1.6 kb/min, for α factor and *cdc7-1* arrest, respectively; Supplementary Fig. 7a), firing time deviation ($\sigma_t$; 2.6 and 9.75 min, respectively; Supplementary Fig. 7b), and collision tract length ($l_{\text{coll}}$; 4.6 kb; Supplementary Fig. 7c, d). In each simulation, firing times were randomly selected for each origin from normal distributions with each origin's listed firing time as the mean and the global $\sigma_t$ as the standard deviation. Given the global $v_f$, and selected firing times, each origin was checked to see if any fork overran it before it could fire. Such origins were removed from that simulation. Fork collision positions were then calculated and collision tracts assigned from that point ± $l_{\text{coll}}$. The density of collision tracts was built up over multiple simulations. Where indicated by RHII-HydEn-seq maps, non-OriDB origin positions ($n = 11$; yellow lines in Supplementary Fig. 3) and firing times ($n = 157$; gray regions in Fig. 1d and Supplementary Fig. 3) were inferred.

The typical simulation run used 1000 iterations near local optimal extrema and 500 iterations elsewhere. The global fork velocity ($v_f$) was optimized first by fitting the RMSD (predicted collision peaks and observed DDAF peaks) vs. velocity curve to a third-order polynomial and finding the local minimum (Supplementary Fig. 7a). Given $v_f$, $l_{\text{coll}}$, and $\sigma_t$ were varied and the fit between the simulation and observed DDAFs was assessed. Resulting optimal $\sigma_t$ vs. $R^2$ curves were fit to third-order polynomials and finding the local maxima found (Fig. 6b). The $\sigma_t$ for the maxima were plotted vs. $l_{\text{coll}}$ and found to have an exponential relationship (Supplementary Fig. 7c; $R^2 = 0.9996$). The optimum $l_{\text{coll}}$ was determined by finding the local maximum for a third-order polynomial fitted to a plot of $l_{\text{coll}}$ vs. maximum $R^2$ (Supplementary Fig. 7d) and the optimum $\sigma_t$ was then calculated using the exponential relationship. Given an optimal $l_{\text{coll}}$ of 4.6 kb, new simulations were run, again varying $\sigma_t$ (Supplementary Fig. 7b, yellow points), to confirm that the predictions from all polynomial and exponential relationships held to within 2% tolerance.

The efficiency of each *S. cerevisiae* origin (Supplementary Data 1) was derived from the proportion of the time that origin was overrun before it fired by forks originating from other origins. Given that the optimal global fork velocity ($v_f$) scaled with replication times ($s_t$), efficiencies were the same for replication times from both α-factor and *cdc7-1* arrest.

**Reporting summary.** Further information on research design is available in the Nature Research Reporting Summary linked to this article.

## Data availability
All sequencing data (raw and mapped) have been deposited in NCBI's Gene Expression Omnibus under the accession number GSE125855.

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

## Acknowledgements

We thank D. Gordenin, S. Williams, N. Degtyareva, and K. Bebenek for critical reading and thoughtful comments on the manuscript. We thank T. Carr for the *S. pombe* strains. We thank High Throughput Genomic Sequencing Facility of UNC Chapel Hill for performing Illumina Hiseq sequencing. We thank Epigenomics Core Laboratory for performing Illumina Miseq sequencing for our pilot projects. This work was supported by Project Z01 ES065070 to T.A.K. from the Division of Intramural Research of the NIH, NIEHS.

## Author contributions

Z.-X.Z., S.A.L. and T.A.K. conceived the project and wrote the manuscript. Z.-X.Z. optimized RHII-HydEn-seq procedures and constructed sequencing libraries. S.A.L., A.B.B., and Z.-X.Z. performed data analysis. M.A.G. tested the restriction enzyme digestion.

## Additional information

**Competing interests:** The authors declare no competing interests.

