## [Peer Review File · Nature Communications]

REVIEWERS' COMMENTS:

Reviewer #1 (Remarks to the Author):

The study is novel and of significant interest to the field.
To make the study clearer, the points shown below should be addressed.

Authors should provide a reference for the L03 reference genome that is mentioned on page 16 of the manuscript.

Authors should provide an explanation regarding why normalization to SbfI sites is necessary since this is a new type of normalization compared to their previous protocol. Previously they only normalized to counts per million mapped reads.

Authors should show position of rNMP relative to the sequenced read in Fig S1. The position of the rNMP is only shown in the first few steps of the figure, but not in the final product sent for sequencing, which is the most important step in terms of data analysis. While it is reasonable that an "R" label could be misleading, still it is important to mark the position of the ribonucleotide, it could be 'D' or some other symbol, as far as it is clear where the ribonucleotide position is. This is relevant for the data analyses.

In regard to the calculation steps for polymerase division, it does not appear straightforward how Eq.6 and Eq.7 are generated from the former equations because a lot of key steps are not shown. It would be helpful to provide more complete details of these equations so that it is possible to reproduce the calculations.

Reviewer #1 (Remarks to the Author):

The study is novel and of significant interest to the field.
To make the study clearer, the points shown below should be addressed.

Authors should provide a reference for the L03 reference genome that is mentioned on page 16 of the manuscript.

We have added the reference.

Authors should provide an explanation regarding why normalization to SbfI sites is necessary since this is a new type of normalization compared to their previous protocol. Previously they only normalized to counts per million mapped reads.

We added the following sentence “SbfI sites were used for normalization instead of total mapped reads because the restriction sites are not influenced by ribonucleotide incorporation and thus are better internal standards.”

Authors should show position of rNMP relative to the sequenced read in Fig S1. The position of the rNMP is only shown in the first few steps of the figure, but not in the final product sent for sequencing, which is the most important step in terms of data analysis. While it is reasonable that an “R” label could be misleading, still it is important to mark the position of the ribonucleotide, it could be ‘D’ or some other symbol, as far as it is clear where the ribonucleotide position is. This is relevant for the data analyses.

We have revised the figures and used an “R” of different color to indicate the position of the ribonucleotide in the sequencing template.

In regard to the calculation steps for polymerase division, it does not appear straightforward how Eq.6 and Eq.7 are generated from the former equations because a lot of key steps are not shown. It would be helpful to provide more complete details of these equations so that it is possible to reproduce the calculations.

In an attempt to be brief, Eq. 6 was used as an example for a class of equations. This was both unclear and unwise. We thank the reviewer for finding this error. The calculation section of the Supplemental Methods has been thus expanded and edited for clarity.